# Terahertz Spectroscopy Characterization and Prediction of the Aging Degree of Polyethylene Pipes Based on PLS

**DOI:** 10.3390/ma16103652

**Published:** 2023-05-11

**Authors:** Jiaojiao Ren, Jisheng Xu, Dandan Zhang, Jiyang Zhang, Lijuan Li

**Affiliations:** 1Key Laboratory of Photoelectric Measurement and Optical Information Transmission Technology of Ministry of Education, Changchun University of Science and Technology, Changchun 130022, China; 2College of Optoelectronic Engineering, Changchun University of Science and Technology, Changchun 130022, China; 3Zhongshan Institute, Changchun University of Science and Technology, Zhongshan 528400, China

**Keywords:** polyethylene, aging prediction, terahertz, spectral slope, partial least square

## Abstract

Polyethylene (PE) is widely used in pipeline transportation owing to its excellent corrosion resistance, good stability, and ease of processing. As organic polymer materials, PE pipes inevitably undergo different degrees of aging during long-term use. In this study, terahertz time-domain spectroscopy was used to study the spectral characteristics of PE pipes with different degrees of photothermal aging, and the variation in the absorption coefficient with aging time was obtained. The absorption coefficient spectrum was extracted using uninformative variable elimination (UVE), successive projections algorithm (SPA), competitive adaptive reweighted sampling (CARS), and random frog RF spectral screening algorithms, and the spectral slope characteristics of the aging-sensitive band were selected as the evaluation indices of the degree of PE aging. Based on this, a partial least squares aging characterization model was established to predict white PE80, white PE100 and black PE100 pipes with different aging degrees. The results showed that the prediction accuracy of the absorption coefficient spectral slope feature prediction model for the aging degree of different types of pipes was greater than 93.16% and the verification set error was within 13.5 h.

## 1. Introduction

Polyethylene (PE) is a type of high polymer material with corrosion resistance, high toughness, long life, and stable properties, and is widely used in medium- and low-pressure gas pipelines, water supply, and drainage pipelines in cities and towns. In the course of its use, PE is often affected by light, temperature, oxygen, humidity, pressure, and other factors, resulting in a decline in the mechanical properties and the polyethylene molecular chain breaks of aging pipes, gas leakage, and accidents, causing serious economic losses [1,2,3]. Currently, methods for characterizing the degree of aging of PE pipes mainly include mechanical property characterization and spectroscopy. In the characterization of mechanical properties, Chen et al. [4] analyzed the change in the nonlinear parameters and tensile properties of PE80 and verified the feasibility of using a nonlinear ultrasonic method to characterize the aging degree of PE pipes. Chen et al. [5] studied the thermal oxidative aging behavior of PE pipes under constant pressure and cyclic pressure at 80, 95, and 110 °C by an accelerated aging test. The tensile test results show that the fracture strength of the PE pipes decreases with an increasing aging time. Simultaneously, the number of pipes aged at higher temperatures decreases faster. Compared with a constant pressure, the cycle pressure decreases faster. The mechanical properties are characterized by destructive experiments, which can only be used as confirmatory experiments. In spectral detection, Hoàng et al. [6] used a combination of a hydrostatic pressure test and chemical analysis to detect the hydrogen peroxide oxidation product (ROOH) of PE100 water pipe material by iodometry, and developed an empirical model based on Arrhenius to infer the service life of a PE100 steel pipe at 10–25 °C. However, infrared spectroscopy has a large error and low sensitivity in quantitative analysis, and mainly relies on experience in spectrum analysis.

At the end of the 20th century with the development of optical devices, spectral detection research has been broadened to the terahertz (THz) field [7]. Chang et al. [8] used THz dielectric-constant spectroscopy to characterize the aging phenomenon of materials, and the concurrent change rate of the dielectric constant significantly changed with a different aging time caused by ultraviolet light. Wang et al. [9] preprocessed the THz optical parameters of materials with a different aging time, and established the aging model by using the second derivative spectral preprocessing method. The optimal coefficient of determination was 99.04%. In 2022, Cheng et al. [10] obtained the absorption spectrum of material aging by THz, and established the theoretical model of the vibration frequency of THz absorption peak, verifying it by the molecular simulation method. The results showed that the change of the molecular group leads to the change of the 1.25 THz absorption peak, and the absorption peak has a negative exponential relationship with aging time. In 2017, Yan et al. [11] used terahertz time-domain spectroscopy (THz-TDS) technology to detect the aging degree of cross-linked polyethylene insulated cables, and found that in the frequency range of 0.7–2.5 THz, the dielectric constant decreases with the increase of aging time. Chen et al. [12] used THz-TDS to detect and analyze artificially aged PE materials. The experimental results showed that the absorption coefficient of the material decreased significantly with the increase of aging time. It is confirmed that terahertz can characterize the aging of polymer materials.

In this study, the characterization and prediction of the photothermal aging degree of PE pipes for outdoor use were studied based on terahertz time-domain spectroscopy [13,14]. Firstly, aging samples of the PE pipes were simulated and manufactured. Secondly, the absorption coefficient spectra of the PE samples that were aged for different durations were extracted and analyzed. Finally, a partial least-squares aging prediction model was established based on the spectral slope of the aging-sensitive band to characterize the aging degree of the PE pipes.

## 2. Materials and Methods

### 2.1. Preparation of Aging Samples

PE pipes used for gas pipes and water pipes are mainly PE80 and PE100, which can maintain the minimum strength of 8.0 MPa and 10.0 MPa after 50 years of use at 20 °C. To better simulate the aging of PE pipes in the practical application of perennial exposure to outdoor use, white PE80, white PE100, and black PE100 pipes were selected for comparative experiments with a length and width of 20 × 100 mm and a thickness of 3 mm. According to the national standard GB/T 16422.2-2022, the PE pipes were subjected to accelerated aging in a xenon lamp aging test chamber. An imported air-cooled Xe arc light source was used in the test chambers. The spectral wavelength ranged from 280 to 800 nm. The experimental conditions were irradiance of 50 W/m^2^, a black standard temperature of 65 ± 3 °C, and a relative humidity of 65% ± 10%. The dry-wet cycle was 102 min of drying and 18 min of spraying [15]. The nozzle aperture is 0.8 mm, and the rain water pressure was between 0.12–0.15 kPa. The PE samples were placed on the rotating disk of the aging box to maintain the aging conditions of each sample were consistent, and 20 groups of PE aging samples with different degrees of aging were prepared at intervals of 12 h. The artificially aged samples are shown in Figure 1.

### 2.2. Terahertz Detection System

The terahertz time-domain spectroscopy system used in this study was an Advantest model TAS7500SP spectrometer, as shown in Figure 2. The broadband measurement capability of a spectrometer enables various spectral analysis applications. The module supports liquid, solid, and powder samples. The frequency range is 0.5–7 THz, and the dynamic range is more than 57 dB. An external air-drying unit was used to eliminate moisture interference during detection. The samples with different aging times were cut and placed in a spectrometer for optical parameter extraction. Each sample was tested 10 times. More accurate optical parameters for different aging times were obtained by averaging ten groups of different optical parameters.

## 3. Results and Discussion

### 3.1. Extraction of Optical Parameters of PE Pipes

Assuming that the thickness of the sample is d, the corresponding THz time-domain reference signal $E_{ref}\left(t\right)$ and sample signal $E_{sam}\left(t\right)$ were obtained, and the time-domain signal was converted to the frequency-domain $E_{ref}\left(\omega \right)$ and $E_{sam}\left(\omega \right)$ by Fourier transform.

According to the Fresnel coefficient, the transfer function of the material is:(1)$$H\left(\omega \right)=\frac{E_{sam}\left(\omega \right)}{E_{ref}\left(\omega \right)}=\rho \left(\omega \right)e^{-jf\left(\omega \right)}$$
where ω is the angular frequency, j is the imaginary unit, ρ(ω), and ϕ(ω) represents the amplitude and phase of the transfer coefficients.

The equations for calculating the refractive index, extinction coefficient, and absorption coefficient of the samples are as follows [16,17]:(2)$$n_{1}\left(\omega \right)=\phi \left(\omega \right)\frac{c}{\omega d}+1$$
(3)$$\kappa_{1}\left(\omega \right)=\ln\left[\frac{4n_{1}\left(\omega \right)}{\rho \left(\omega \right){\left[1+n_{1}\left(\omega \right)\right]}^{2}}\right]\frac{c}{\omega d}$$
(4)$$\alpha_{1}\left(\omega \right)=\frac{2\kappa_{1}\left(\omega \right)\omega }{C}=\frac{2}{d}\ln\left[\frac{4n_{1}\left(\omega \right)}{\rho \left(\omega \right){\left[1+n_{1}\left(\omega \right)\right]}^{2}}\right]$$

### 3.2. Spectral Characteristics Analysis of PE pipes

Equation (4) was used to calculate the absorption coefficients of the three types of PE pipe samples with different aging times. Because of the internal electrical noise and background stray light of the spectrometer, the absorption coefficient was SG-smoothed before imaging analysis to eliminate the influence of curve fluctuations on the results in the selected interval [18]. Considering the influence of the signal-to-noise ratio of the THz wave energy on the overall data quality and the sensitive range of the PE pipe to the THz spectrum, a pulse width range of 1.5–4 THz was selected, and the curve of the absorption coefficient with aging time was obtained, as shown in Figure 3. The absorption coefficient of three types of different aging time had an absorption peak of approximately 2.2 THz [19]. The absorption peak of white PE100 and white PE80 was approximately 2 cm^−1^, and the absorption peak of black PE100 was larger, approximately 4 cm^−1^, indicating that the density of the PE pipe had little effect on the absorption coefficient, and the effect was greater after adding carbon black masterbatch. The frequency domain waveform after the absorption peak decreased with an increasing aging time, and the absorption coefficients of some aging times overlapped. In the same artificial aging environment, the absorption coefficient of white PE80 was more evident than that of white PE100, indicating that the aging resistance of white PE100 was stronger owing to the high density of PE. The absorption coefficient of black PE100 was significantly lower than that of white PE100, confirming that black PE100 containing a carbon black masterbatch can effectively prevent the PE molecular chain from being damaged by ultraviolet light and is not easy to age.

### 3.3. Spectral Feature Extraction

The influence of more THz spectral data band points and a large amount of data may affect the subsequent modeling and prediction effects. Wavelength screening is typically used to eliminate redundant information in the THz spectrum to extract effective features, thereby improving the modeling efficiency and reducing the amount of calculation. In this study, four spectral band screening methods—uninformative variable elimination (UVE) [20], successive projections algorithm (SPA) [21], competitive adaptive reweighted sampling (CARS) [22] and random frog (RF) [23]—were used to extract the characteristic wave of the THz absorption coefficient of three PE pipes.

The spectral sensitivity range of the PE pipe was 1.5–4 THz, resolution was 0.0075 THz, and number of bands was 329. As shown in Figure 4, the absorption coefficient of white PE100 was extracted using four spectral bands. The number of characteristic wavelengths selected by the four methods was greatly reduced compared to the number of original wavelengths, and four wavelength screening methods were used to screen white PE80 and black PE100 materials. The wavelengths selected for the three materials are listed in Table 1. The wavelengths of the white PE80 decreased to 15.8%, 5.47%, 4.25% and 5.77% of the full spectrum, respectively. The wavelengths of the white PE100 decreased to 13.06%, 6.07%, 3.64%, and 3.64% of the full spectrum. The wavelengths for black PE100 decreased to 10.33%, 4.25%, 2.43%, and 5.47% of the full spectrum.

Most of the wavelengths of the three types of pipes after screening were in the range of 2.5–4 THz. The wavelengths of UVE, SPA, CARS, and RF of white PE100 were in the range of 2.5–4 THz and accounted for 86%, 90%, 83%, and 75%, respectively, and the wavelengths of the other two PE pipes accounted for more than 75%. Therefore, this paper used polynomial fitting method to fit the absorption coefficient curves of three PE pipes in the range of 2.5–4 THz with 0.25 THz as the interval. The slope change curve after fitting is shown in Figure 5 [24].

The slope of the three PE pipes decreased with the increase of aging time in the range of 2.5–4 THz, and the slope of the aging absorption coefficient of white PE80 in 2.5–2.75 THz and 2.75–3 THz was linear with the aging time. The aging absorption coefficient slope of white PE100 at 2.5–3 THz and 3–3.25 THz was linear with the aging time. The slope of aging absorption coefficient of black PE1000 in 2.5–2.75 THz, 2.75–3 THz and 3–3.25 THz was linear with aging time. The proportions of the filtered wavelengths in the linear range are shown in Table 2.

It can be seen that the average wavelength of 64.7% after spectral-feature band screening coincided with the linear relationship interval of the spectral slope. A band with a linear relationship interval of the spectral slope can be defined as an aging-sensitive band, and the feasibility of the spectral slope feature can be further verified by partial least-squares modeling.

### 3.4. Dataset Description

In this paper, two sets of aging experiments are carried out for three kinds of aging samples to form a data set, one set of data is used for training verification, one set of data is tested, and the specific data set is composed as shown in Figure 6. The training verification data are 21 × 3, the number of unaged sample data are 1, and the rest comprises 20 sample data with an aging time interval of 12 h. Additionally, 70 % of the training set and 30 % of the test sets are randomly selected. The test data are sample data with an aging time of 50 h, 100 h, 150 h and 200 h, respectively.

In order to avoid the influence of the detection environment and the detection equipment on the spectral results, this paper selects all samples to carry out centralized detection at the same time after the aging experiment, so as to ensure the consistency of the detection data and the spectral accuracy.

### 3.5. Aging Model Establishment and Evaluation

Partial least-squares (PLS) regression is a new multivariate statistical analysis method that integrates multiple linear regressions, canonical correlations, and a principal component analysis [25]. Supposing that the number of independent variables in the study sample is p, the number of observations of each variable is *n*, the independent variable matrix is denoted by X, and the dependent variable matrix is denoted by Y, then let t be the number of principal components that must be extracted in X, and when X extracts the principal component $t_{1}$, the following two conditions must be satisfied:

(1)$t_{1}$ should extract the variation information of the respective data table variables as much as possible; (2)The correlation between $t_{1}$ and Y is as high as possible.

If the regression effect meets the preset requirements, the PLS regression process is terminated; otherwise, it is necessary to use the information remaining after X is interpreted by $t_{1}$ to continue screening for the second principal component, $t_{2}$. These steps were repeated until the final preset requirements were met [26].

In this study, the correlation coefficient (*R*) and root mean square error (*RMSE*) were used as evaluation indices for the model performance. *R* was used to measure the degree of correlation of the model, and *RMSE* was used to evaluate the quality and predict the ability of the quantitative analysis model. The calculation equations are as follows [27]:(5)$$R=\sqrt{1-\frac{\sum_{i=1}^{m}{\left(y_{i}-f_{i}\right)}^{2}}{\sum_{i=1}^{m}{\left(y_{i}-{\bar{y}}_{i}\right)}^{2}}}$$
(6)$$RMSE=\sqrt{\frac{\sum_{i=1}^{m}{\left(y_{i}-f_{i}\right)}^{2}}{m}}$$

In the above equation, m is the number of samples, $y_{i}$ is the reference value of the samples, and $f_{i}$ is the predicted value of the samples, where ${\bar{y}}_{i}$ represents the average reference value of the m sample. The closer *R* is to 1 and *RMSE* is to 0, the better the prediction performance of the model [28].

The slope of the aging-sensitive band was selected as the feature for PLS modeling, and the feasibility of the spectral slope feature for predicting the degree of aging of the PE pipe was verified based on the modeling results. The modeling effects are listed in Table 3.

It can be seen from Table 3 that the slope of the aging-sensitive band is selected as the feature compared with the original spectrum, which not only simplifies the model but also improves accuracy. It was confirmed that the selected band was sensitive to the aging degree of the PE pipe; therefore, the slope of the aging-sensitive band was used as the modeling feature of the aging model to predict the aging degree.

### 3.6. Aging Prediction Results and Analysis

The PLS aging prediction model was established using the spectral slope as the spectral feature, and the absorption coefficient data of 20 groups of aging samples (white PE80, white PE100 and black PE100) were predicted. As shown in Figure 7, the training, verification results, and errors of the PE pipes with three different degrees of aging were characterized by the slope of the absorption coefficient. 

The accuracies of the white PE80 training set and verification sets were 3.38% and 5.62%, respectively. The accuracies of the white PE100 training and verification set were 3.27% and 5.73%, respectively. The accuracies of the black PE100 training and verification set were 3.14% and 6.84%, respectively. The training set errors for the three materials were less than 9.8 h, whereas the verification set errors were less than 13.2 h. The error of the three material test sets is shown in Table 4.

Among them, the minimum error of the white PE80 test sets is 4.08 h. The minimum error of the white PE100 prediction sets is 5.9 h. The minimum error of the black PE100 prediction sets is 3.9 h, and the error in the three pipe test sets were within 13.5 h. It was confirmed that the THz spectral characteristics can be used to characterize and predict the aging degree of PE pipes.

## 4. Conclusions

In this study, the characterization and prediction of the degree of photothermal aging of PE pipes for outdoor use were studied based on terahertz time-domain spectroscopy. Firstly, the terahertz spectrometer was used to extract the optical parameters of the artificial white PE80, white PE100 and black PE100 photothermally aged samples, and the absorption coefficient spectrum was analyzed. With increasing aging time, the absorption coefficient spectra of the three materials show a downward trend. Secondly, the UVE, SPA, CARS, and RF spectral screening algorithms were used to screen the wavelength characteristics of the absorption coefficient spectrum, and the spectral slope characteristics of the aging-sensitive band were selected as an evaluation index of the degree of PE aging. Finally, a PLS aging characterization model was established to characterize and predict the three types of PE pipes with different degrees of aging. The *RMSEc* and *RMSEcv* of the model for the three pipes were less than 0.0615 and 0.0753, respectively. *Rc* and *Rcv* were above 0.98 and 0.966, respectively. The prediction accuracy of three different aging degrees of white PE80, white PE100 and black PE100 was 94.38%, 94.27% and 93.16% verification set error was within 13.5 h. The slope of the absorption spectrum not only simplified the model but also improved the prediction accuracy, proving that the optical parameter characteristics of the THz band can be used to characterize the degree of aging of PE pipes.

## Figures and Tables

**Figure 1 materials-16-03652-f001:**
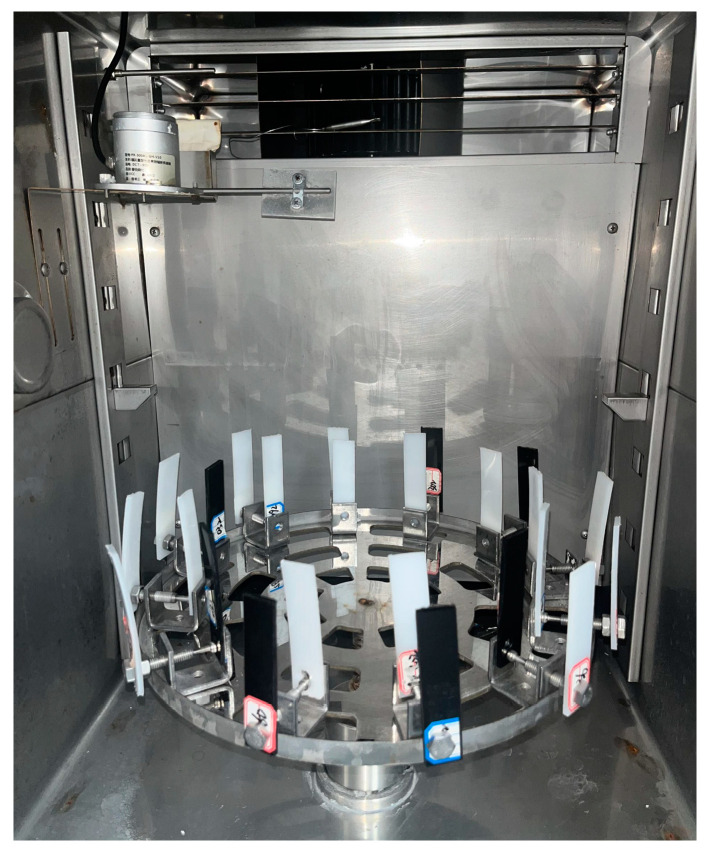
Artificial aging sample diagram.

**Figure 2 materials-16-03652-f002:**
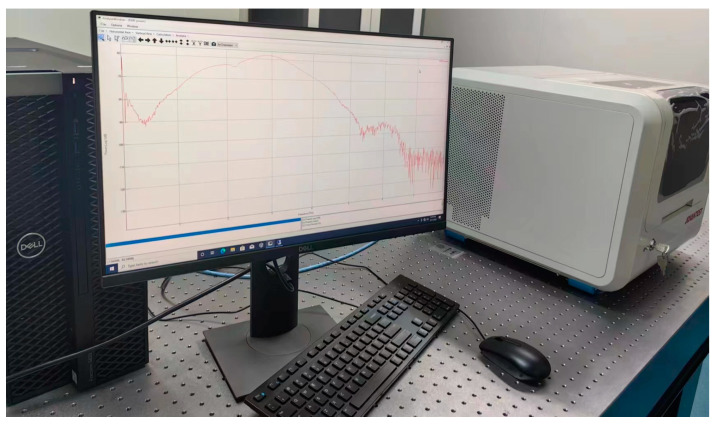
THz spectrometer.

**Figure 3 materials-16-03652-f003:**
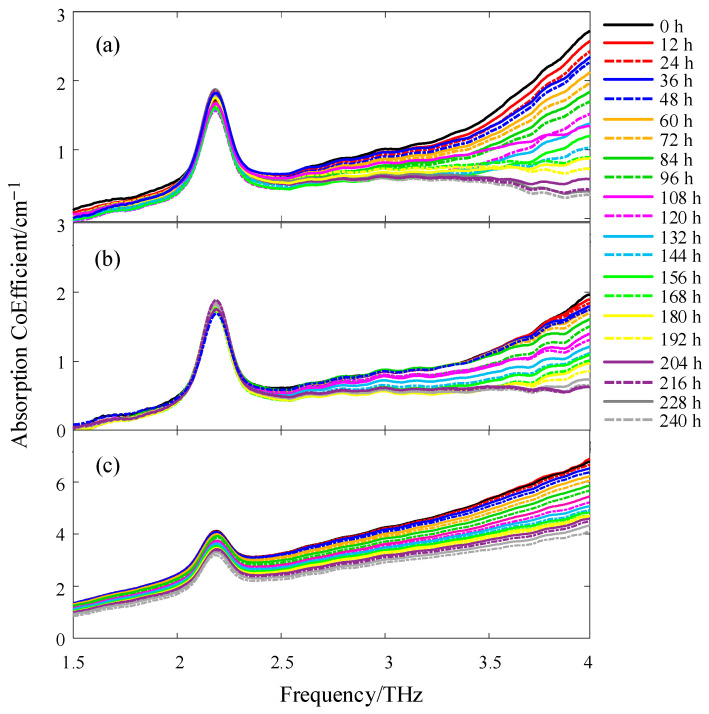
Absorption coefficients of three types of different aging time: (**a**) White PE80; (**b**) White PE100; (**c**) Black PE100.

**Figure 4 materials-16-03652-f004:**
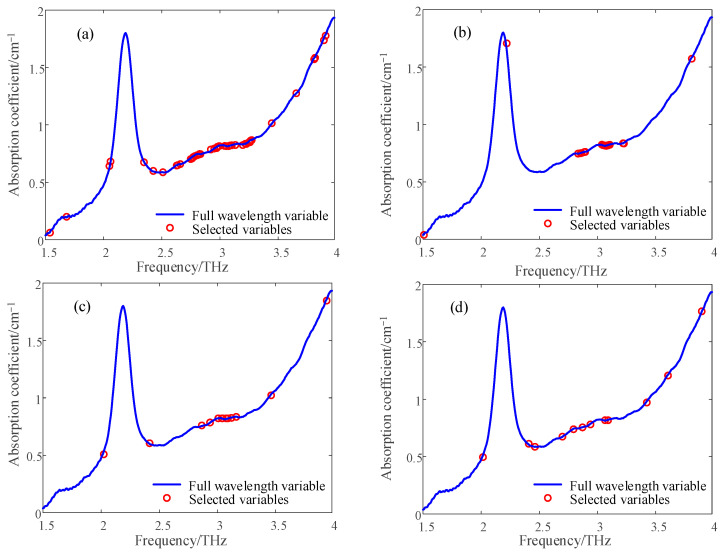
White PE100 terahertz absorption coefficient spectral band extraction results: (**a**) UVE; (**b**) SPA; (**c**) CARS; (**d**) RF.

**Figure 5 materials-16-03652-f005:**
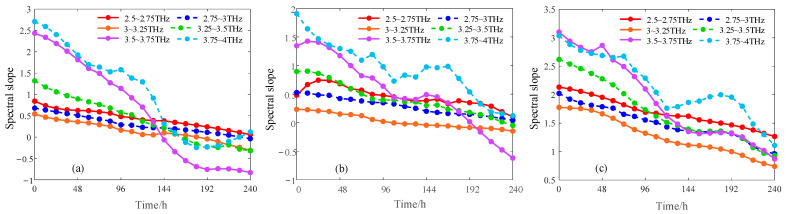
Three PE pipes in the range of 2.5–4 THz slope with the aging time curve: (**a**) White PE80; (**b**) White PE100; (**c**) Black PE100.

**Figure 6 materials-16-03652-f006:**
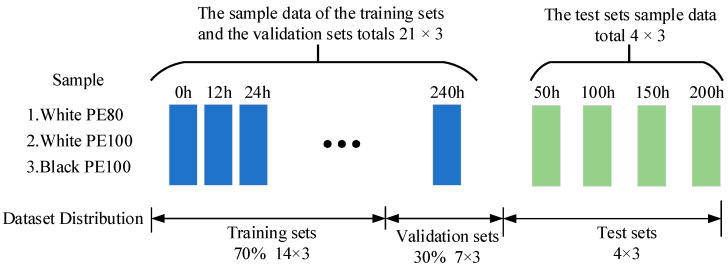
Data set composition.

**Figure 7 materials-16-03652-f007:**
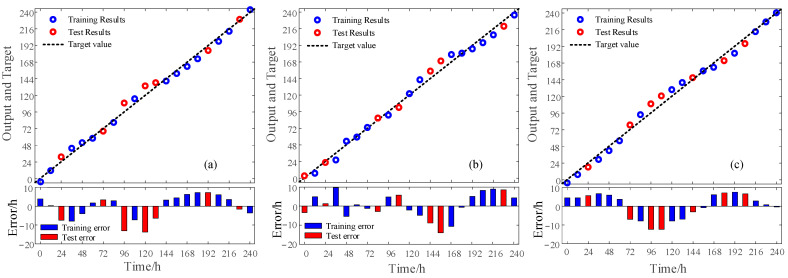
The training, verification results and errors of three different aging degree PE pipes in 0–240 h: (**a**) White PE80; (**b**) White PE100; (**c**) Black PE100.

**Table 1 materials-16-03652-t001:** The number of wavelengths after spectral band screening of three materials.

PE Pipe Types	UVE	SPA	CARS	RF
White PE80	52	18	14	19
White PE100	43	20	12	12
Black PE100	34	14	9	18

**Table 2 materials-16-03652-t002:** Proportion of the wavelengths of the three PE pipes after screening in the linear relationship range.

Sieving Algorithm	White PE80 Spectral Range/THz		White PE100 Spectral Range/THz		Black PE1100 Spectral Range/THz		
	2.5~2.75	2.75~3	2.75~3	3~3.25	2.5~2.75	2.75~3	3~3.25
UVE	0.63		0.65		0.62		
SPA	0.72		0.75		0.71		
CARS	0.71		0.67		0.67		
RF	0.58		0.50		0.55		

**Table 3 materials-16-03652-t003:** Slope feature modeling effect.

Feature Input	*Rc*	*RMSEc*	*Rcv*	*RMSEcv*
White PE80 Full Spectrum	0.964	0.0732	0.957	0.0821
White PE100 Full Spectrum	0.976	0.0765	0.959	0.0835
Black PE100 Full Spectrum	0.972	0.0674	0.962	0.0996
Spectral slope characteristics of white PE80	0.98	0.0615	0.968	0.0753
Spectral slope characteristics of white PE100	0.982	0.0693	0.966	0.0711
Spectral slope characteristics of black PE100	0.989	0.0498	0.975	0.0596

**Table 4 materials-16-03652-t004:** Three kinds of material test set error.

Feature Input	50 h Error/h	100 h Error/h	150 h Error/h	200 h Error/h
White PE80	6.62	4.08	10.66	9.78
White PE100	8.43	9.94	5.9	10.8
Black PE100	6.95	3.9	8.5	13.5

## Data Availability

The data used to support the findings of this study are available from the corresponding author upon request.

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
