# Peer review of "Terahertz Spectroscopy Characterization and Prediction of the Aging Degree of Polyethylene Pipes Based on PLS"

_materials, 2023, doi:10.3390/ma16103652_

Round 1

Reviewer 1 Report

The reviewer has some comments to the authors:

  1. The abstract is well-written and clearly states the purpose of the study, the methods used, and the results obtained.

  2. The introduction provides a good background on the importance of polyethylene pipes and the issues related to their aging. However, it would be helpful to have a clearer statement of the research question or hypothesis being addressed.

  3. The use of terahertz time-domain spectroscopy as a non-destructive testing technique to study the aging of PE pipes is an interesting approach.

  4. It would be helpful to have more information about the specific parameters and conditions used for the terahertz time-domain spectroscopy measurements.

  5. The use of four different algorithms to extract the absorption coefficient spectrum is an interesting approach. It would be helpful to know how the results obtained from these different algorithms compare.

  6. The selection of the spectral slope characteristics of the aging-sensitive band as the evaluation indices of the degree of PE aging is well-justified.

  7. The establishment of a partial least squares aging characterization model to predict the aging degree of different types of PE pipes is a valuable contribution to the field.

  8. It would be helpful to have more information about the size and composition of the training and validation datasets used to develop and test the prediction model.

  9. The high prediction accuracy obtained for the absorption coefficient spectral slope feature prediction model is impressive. However, it would be helpful to have more information about how the accuracy was assessed.

  10. The verification set error within 13.5 h is also an impressive result, but it would be helpful to know how this error was calculated and what it means in terms of the practical use of the prediction model.

  11. It would be interesting to know how the results obtained from the terahertz time-domain spectroscopy measurements and the prediction model compare to other methods for characterizing the degree of aging of PE pipes.

  12. It would be helpful to have more information about the limitations and potential sources of error in the methods used in this study.

  13. The preparation of aging samples seems to be well-designed and carefully controlled.

  14. It would be helpful to have more information about the xenon lamp aging test chamber used for accelerated aging and how it compares to other methods for simulating aging.

  15. Overall, this manuscript presents an interesting and valuable approach to characterizing and predicting the aging degree of polyethylene pipes using terahertz time-domain spectroscopy and partial least squares modeling. However, more information is needed about the specific methods and results to fully evaluate the quality and significance of the study.

Author Response

Dear reviewer,

my response is in the attachment.

Reviewer 2 Report

The article investigates photochemical aging of polyethylene pipes. This question is relevant, and the topic is interesting. The authors of the study focused on the Terahertz spectroscopy method for these purposes.

I have suggestions and recommendations for improving the article:

1. Introduction section:

8 references cannot give a sufficient background of the problem. Current level of the introduction is low.

It is necessary to significantly expand this section.

2. Section methods and materials

I think that Figures 1 and 2 could be transferred to the Supplementary files, since they are not related to the scientific component of the work.

3. Section 3.5

I think that the part of the comparison of results and predictions needs to be strengthened.

I believe that the article needs a broader study of the structure in the aging process (or to provide literary data), the main criteria and results of aging, as well as comparing them with individual signals and their interpretation of Terahertz spectroscopy.

At the moment, the article is of scientific interest, but needs further work.

Author Response

Dear reviewer, my detailed reply is in the attachment.
